Subject Area:
developmental biology

Keywords:
vertebrates, epithelial–mesenchymal transition, lamprey, neural crest, evolution, chordates

Author for correspondence:
David W. McCauley
e-mail: dwmccauley@ou.edu

# The origin and evolution of vertebrate neural crest cells

Joshua R. York and David W. McCauley

Department of Biology, University of Oklahoma, 730 Van Vleet Oval, Norman, OK 73019, USA

 JRY, 0000-0001-8853-4880; DWM, 0000-0002-4190-9304

The neural crest is a vertebrate-specific migratory stem cell population that generates a remarkably diverse set of cell types and structures. Because many of the morphological, physiological and behavioural novelties of vertebrates are derived from neural crest cells, it is thought that the origin of this cell population was an important milestone in early vertebrate history. An outstanding question in the field of vertebrate evolutionary-developmental biology (evo-devo) is how this cell type evolved in ancestral vertebrates. In this review, we briefly summarize neural crest developmental genetics in vertebrates, focusing in particular on the gene regulatory interactions instructing their early formation within and migration from the dorsal neural tube. We then discuss how studies searching for homologues of neural crest cells in invertebrate chordates led to the discovery of neural crest-like cells in tunicates and the potential implications this has for tracing the pre-vertebrate origins of the neural crest population. Finally, we synthesize this information to propose a model to explain the origin of neural crest cells. We suggest that at least some of the regulatory components of early stages of neural crest development long pre-date vertebrate origins, perhaps dating back to the last common bilaterian ancestor. These components, originally directing neuroectodermal patterning and cell migration, served as a gene regulatory 'scaffold' upon which neural crest-like cells with limited migration and potency evolved in the last common ancestor of tunicates and vertebrates. Finally, the acquisition of regulatory programmes controlling multipotency and long-range, directed migration led to the transition from neural crest-like cells in invertebrate chordates to multipotent migratory neural crest in the first vertebrates.

## 1. Introduction

The origin of the vertebrates some 500 Ma was a milestone in early animal evolution that has since led to the diversification of over 69 000 extant species as well as countless others described from the fossil record [1]. Vertebrates have colonized a wide range of ecological niches on every continent, ranging in size from a few millimetres to over 30 m in length [1,2]. However, despite the great diversity and disparity in form and function of adult forms, all vertebrates still share a common set of phenotypic traits, including a genetic blueprint that guides construction of their body plans during embryonic development and reflects their shared ancestry [3–6].

One particularly important embryological feature that all vertebrates share is the neural crest [7–10] (figure 1). Neural crest cells form in the dorsal-most part of the nascent embryonic central nervous system (CNS), from which they detach and then migrate throughout the embryo to give rise to a diverse array of cell types that go on to make up many of the morphological and physiological traits that characterize the vertebrate clade, including most of the craniofacial skeleton and peripheral sensory nervous system, striking patterns of pigmentation, components of the teeth, heart and endocrine system, and much more [9–11] (figures 1 and 2). Because most of these traits are hallmarks

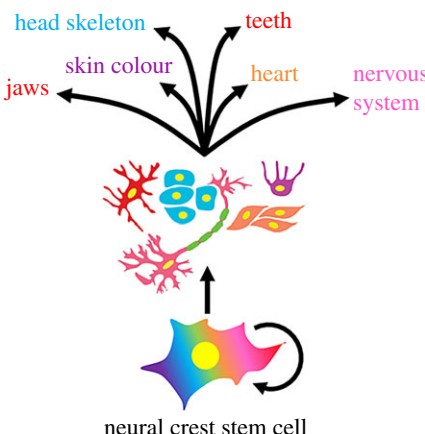

**Figure 1.** Cartoon schematic of a neural crest cell which has stem cell properties (capable of self-renewal, circular arrow) and is multipotent by generating diverse cell types that make up numerous vertebrate structures and tissues.

of the vertebrate body plan, the origin of the neural crest is thought to have been a seminal event in early vertebrate history because it enabled a series of evolutionary transitions that distinguished the vertebrates from their invertebrate chordate relatives [12–16].

Given the importance of neural crest cells to vertebrate development and evolution, studies over the past 30 years have been focused on identifying the ground state for neural crest developmental genetics in early vertebrates, as well as potential homologues of the neural crest among the closest relatives of the vertebrates: the invertebrate chordates. In this review, we summarize briefly the developmental genetics of migratory neural crest stem cells in vertebrates and then use this information within a comparative framework to trace the origins of neural crest-like cells in invertebrates and to build a model to account for the stepwise evolution of regulatory mechanisms driving the production of migratory and multipotent neural crest cells in the first vertebrates.

## 1.1. Neural crest developmental genetics in vertebrates: a primer

At the molecular and genetic levels of organization, neural crest development follows a trajectory that is similar across vertebrates. During or shortly after gastrulation, intercellular signalling via neural crest inducers such as Bmp, Wnt, Fgf and Delta-Notch from the neural plate, epidermal ectoderm and mesoderm [17–19] establishes on either side of the neural plate a zone known as the neural plate border (figure 2). This border region is defined by expression of neural plate border specifier genes such as Zic1, Dlx5, Msx1/2, Pax3/7 and Prdm1 [20–23]. These in turn activate a suite of transcription factors in the dorsal neural tube including SoxEs (Sox 8/9/10), Tfap2α, Id, Snail1/Snail2, Myc, Twist, Ets and many others, which segregate neural crest cells in the dorsal neural tube from the underlying neuroepithelium and gives these cells their multipotent, stem cell state [24–27]. Thus, it is the combinatorial expression of these genes that endows the neural crest with a unique 'molecular anatomy' that distinguishes this stem cell population from the rest of the embryo.

Shortly after specification, neural crest cells engage in one of their most striking behaviours—the ability to delaminate from the dorsal neural tube, undergo an epithelial–mesenchymal transition (EMT), and initiate and sustain long-range migration throughout the embryo as a multipotent population capable of generating diverse cell types [28–34] (figure 2). The initiation of migration occurs via signalling inputs from neural crest inducers such as Bmp and Wnts, which activate expression of a large suite of transcription factors, including, but not limited to, SoxE and SoxD group genes [35,36], FoxD3 [37,38], Snail1/Snail2 [39–41], Twist [42–44], Sip1 [45,46], Zeb1 [47,48], LMO4 [49,50] and E12/E47 [51–53]. Many of the regulatory targets of these factors include genes whose products are directly responsible for modulating the ability of neural crest cells to adhere and/or undergo delamination from neighbouring cells [54–57]. For example, Snail1 and Snail2 directly repress epithelial gene batteries, including type I and type II cadherins by binding to E-box (CANNTG) elements at target gene promoters [41,58–60], often with cofactors such as histone deacetylases [61] and transcription factors such as LMO4 [62], Sox9 [63,64] and LIM homeodomain proteins [65]. Another key feature of migratory neural crest is the dynamic regulation of the cellular cytoskeleton accompanied by breakdown of the basal lamina of the neural tube by proteases such as ADAMs and matrix metalloproteases (MMPs) [66–74]. During this period, neural crest cells alter their cell polarity and generate a leading edge for long-range and directed migration by genes such as the Rho family of small GTPases [56,57,75–81].

The total set of gene regulatory interactions described above comprises a logical gene regulatory network (GRN) for neural crest development [21,24–26]. Despite some inter-species variation, studies across vertebrates, including both jawed and jawless vertebrates (lampreys and hagfish), nonetheless suggest that the neural crest GRN is a core feature of vertebrate development that dates back to their last common ancestor [24,27,82–85]. Given that neural crest cells are a novelty of vertebrates, one of the goals in the field of neural crest evolutionary and developmental biology has been to identify the developmental genetics underpinning their origin and evolution in early vertebrates as well as potential neural crest homologues in the closest extant relatives of vertebrates, the invertebrate chordates. In the following sections, we summarize how developmental studies on these animals have influenced views on neural crest developmental evolution.

## 2. Insights from invertebrates into the evolution of neural crest cells

### 2.1. Cephalochordates (amphioxus)

The extant chordate relatives of vertebrates include the cephalochordates (i.e. amphioxus) and the urochordates (also known as tunicates). Comparative embryology studies of these groups have been an important focus of research for those interested in tracing the ancestry of the vertebrates and neural crest cells [86–90]. For most of the latter twentieth century, it had been suggested that the cephalochordates were the sister group to vertebrates, with tunicates as outgroup [91,92]. This phylogenetic framework strongly

royalsocietypublishing.org/journal/rsob  Open Biol. **10**: 190285

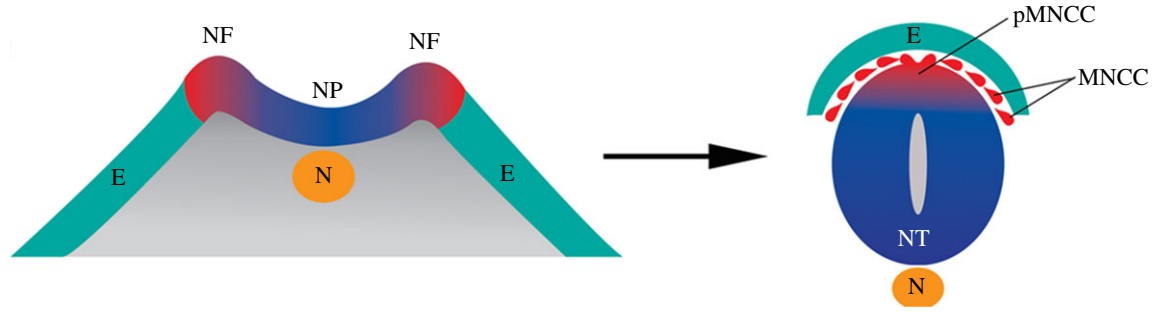

**Figure 2.** General model of neural crest development in a vertebrate embryo. Shown on the left is a cross-section of an open neural plate-stage embryo with neural plate (NP, CNS primordium) in the middle flanked bilaterally by the neural plate borders, which elevate as neural folds (NF) and ventrally by the notochord (N). The epidermal ectoderm (E), presumptive skin is shown extending ventral and lateral to the NPB. Shown on the right is a cross-section of the neural tube (NT) with premigratory neural crest cells (pMNCC) dorsally and migratory neural crest cells (MNCC) exiting this region. Dorsal is up and ventral is down.

influenced hypotheses and interpretations on the origin of vertebrates and neural crest cells [91,92]. More recent molecular phylogenetic studies, however, have flipped this framework on its head. The cephalochordates now reside as outgroup to a vertebrate + tunicate sister group (olfactores), a relationship that is bolstered by the presence of neural crest-like and placode-like cells in tunicates (described in §2.2) [93,94]. In amphioxus, however, there are no cells that have been identified as homologous to neural crest. Genomic analyses have corroborated this by showing that although the amphioxus genome encodes many of the same (single copy) neural crest factors as found in vertebrates, most are not co-expressed in the neural plate border or dorsal neural tube [95,96].

Given the lack of any neural crest cell homologues in amphioxus, several hypotheses have been proposed to explain how changes in gene number, gene regulation or a combination of these two phenomena might have enabled the evolution of neural crest cells during the chordate-to-vertebrate transition. Gene and/or whole-genome duplication has been proposed as a driving force for the evolution of novel gene functions during embryonic development [97–103]. Following the duplication event, one of at least three fates awaits each duplicate. First, one duplicate may acquire deleterious mutations and become non-functional (non-functionalization), with the other copy retaining the ancestral condition [103]. Second, after acquiring mutations, each duplicate may perform only a subset (duplicate 1 = function 'A' and duplicate 2 = function 'B') of the total functions performed by the ancestral gene (ancestral pre-duplicate = functions 'A + B') (subfunctionalization) [103,104]. Third, one duplicate may continue to perform the ancestral function with the other copy acquiring mutations that enable the evolution of novel functional properties (neofunctionalization) [105]. More recently, a fourth potential result of gene duplication has been described in which paralogues may cooperate to perform regulatory functions in ways that are not achieved by single copies [106].

Because neofunctionalization confers novel protein functions shaped by natural selection, it is thought to be a particularly potent mechanism for evolutionary change in GRNs and the acquisition of novel cellular functions. A handful of studies have tested the importance of neofunctionalization in the evolution of neural crest cells. Studies on SoxE transcription factors have found that single-copy SoxE genes in invertebrates such as amphioxus (AmphiSoxE) and *Drosophila* (Sox100B) can ectopically induce migratory neural crest or rescue neural crest defects [107,108]. Similarly, forced

expression of amphioxus Tfap2a or *Drosophila* AP2 in Tfap2a/c-depleted zebrafish rescued several neural crest defects [109,110]. These results highlight that a single, 'pre-duplicate' invertebrate gene can perform all or most of the functions controlled by each duplicate in vertebrates. In contrast with these examples, analysis of FoxD3 function revealed that AmphiFoxD was unable to ectopically produce migratory neural crest in chick embryos [111]. Using gene fusion experiments, the authors of that study traced the neural crest-inducing capacity of chick FoxD3 to a unique string of amino acids that evolved in the amniote lineage [111]. Taken together, these results suggest that, although some novel features of neural crest development and migration may be attributable to duplication and neofunctionalization, there is also evidence that single-copy invertebrate homologues can compensate for the functions of duplicated paralogues in vertebrates. This latter point argues that duplication and specialization of regulatory genes was probably not the main driving force in the evolution of migratory neural crest.

Another important mechanism for developmental evolution involves changes in *cis*-regulatory sequences that direct expression of the associated gene in new cells and tissues (*cis*-regulatory evolution). Is there evidence that *cis*-regulatory evolution played an important role in the origin of neural crest cells? To test this idea, researchers have isolated amphioxus *cis*-regulatory elements for homologues of FoxD3 and SoxE genes, and tested their ability to mediate reporter gene expression in vertebrate embryos. If amphioxus elements can drive reporter expression in vertebrate neural crest cells, then this would mean that these elements pre-date vertebrate origins and are therefore unlikely to be causal to the evolution of the neural crest. By contrast, if no reporter expression is observed in the neural crest, then this indicates that the amphioxus element lacks the full regulatory information required to mediate proper expression and that *cis*-regulatory evolution in vertebrates was required for directing expression of the associated gene in the neural crest domain. In both cases, the amphioxus FoxD and SoxE elements drove expression in non-neural crest-derived tissues (e.g. mesoderm, somites) [112,113]. However, there was no reporter expression observed in premigratory or migratory crest cells [112,113]. These findings suggest that the amphioxus elements lack the regulatory sites to mediate expression in the neural crest, which probably evolved in early vertebrates. Recent comparisons of whole-genome regulatory landscapes between vertebrates and amphioxus have arrived at similar conclusions [114].

Vertebrate genomes have acquired many new enhancers that in turn have enabled greater specialization and precision in spatial–temporal gene expression compared to the ancestral chordate condition [114]. Thus, much of the complexity of *cis*-regulatory control in vertebrate genomes in general may be attributable largely to gene and/or genome duplication, though the extent to which this can be linked to neural crest evolution may be disputed (see §3.2). This increase in overall regulatory complexity may have been possible because duplication events would have allowed a subset of retained paralogues to acquire novel enhancers, while others would have been able to still perform the ancestral regulatory function(s).

## 2.2. Tunicates

Although cephalochordates do not have any migratory cells that can be homologized with vertebrate neural crest cells, tunicates have a couple of different cell types that are strikingly similar to neural crest in several ways. The first of these discoveries came with cell lineage tracing experiments using the lipophilic vital dye, DiI [115–117]. Jeffery *et al.* showed that a species of tunicate (*E. turbinata*) possessed cells that migrated as small streams from the neural tube similar to neural crest cells and gave rise to pigment—a known neural crest derivative—in the body wall and developing siphons of the larva. These 'neural crest-like cells' also expressed neural crest markers such as Zic and HNK1 [115–117]. Subsequent studies revealed the expression of additional neural crest regulatory genes in the a7.6 lineage [115,116]. However, there are also subtle differences in these cells across tunicates and between vertebrates and tunicates. Most notably, in *Ciona* species, this neural crest-like population occupies a relatively small portion of the developing neural plate border and neural tube compared to vertebrates [115–117].

A second neural crest-like population was described in the tunicate, *C. intestinalis*. These cells originate from the a9.49 lineage in the tadpole head, express a neural crest regulatory 'signature' (*Msx*, *Pax3/7*, *Zic*, *Id*, *Snail*, *Ets*, *FoxD*) and migrate a short distance from their site of origin before differentiating into sensory pigment cells of the otolith and ocellus [118]. Although the distance that these cells migrate is quite short, forced expression of *Twist* induces long-range migration into the tunic in a pattern reminiscent of migratory crest in vertebrates [118].

The most recent discovery of neural crest-like cells in tunicates is that of bipolar tail neurons (BTNs) in the larval trunk [119]. BTNs have several characteristics that suggest an affinity with neural crest, including expression of Snail, Msx, Pax3/7 and Zic in the neural plate border, and migration along paraxial mesoderm to their final destinations [119]. Additionally, BTNs are similar to a known neural crest derivative: dorsal root sensory ganglia (DRG). Differentiated BTNs and DRGs both express Neurogenin and Islet and share developmental, morphological and functional similarities. There is also evidence that BTN precursor migration depends on differential regulation of intercellular adhesion proteins similar to delamination and EMT of neural crest cells [119]. The authors found that whereas the epithelial neural tube expresses Cadherin-b, migrating BTNs do not. Conversely, forced expression of Protocadherin-c prevented delamination and migration of BTNs. All of this provides strong evidence

that tunicates possess cells that have the molecular, cellular and genetic hallmarks of neural crest and suggests that a homologous cell population to the neural crest can be found among invertebrate chordates [119].

# 3. Putting it all together: the emergence of neural crest cells

## 3.1. Ancient origins of neural crest regulatory mechanisms

What makes vertebrate neural crest cells and their developmental trajectory unique from other cell types? An adequate answer to this question has become elusive, given the discovery of neural crest-like cells in invertebrate chordates. What these studies have revealed is that many of the molecular and cellular features thought to be unique to the neural crest have deeper evolutionary roots among chordates. However, it is increasingly likely that some of these features extend far beyond even the chordates into early bilaterian history.

Take, for example, the neural plate border in vertebrates, the embryonic domain that produces neural crest progenitors. Studies of invertebrates on both the protostome and deuterostome sides of the bilaterian tree have revealed the presence of so-called lateral neural borders that are similar to the neural plate border [120] (figure 3*a*). These lateral neural borders develop as part of a broad embryonic domain that instructs medial–lateral patterning of the neuroectoderm into the CNS and PNS [120]. Cells derived from lateral neural borders express homologues of Pax, Zic, Msx and Nkx transcription factors and give rise to migratory and non-migratory sensory neurons of the embryonic PNS, just as neural crest cells do in vertebrates [120,121]. A similar situation occurs in tunicates in which neural crest-like cells migrate from the neural plate border and form sensory neurons (BTNs). Thus, a lateral neural border region defined minimally by combinatorial expression of Msx, Zic and Pax transcription factors and production of PNS sensory neurons may be a shared feature of bilaterians that long pre-dates neural crest and vertebrate origins. Although this alone does not prescribe strict homology with vertebrate neural crest cells, it does suggest that lateral borders and their underlying GRNs may be homologous across bilaterians [120,121]. This would mean that the neural plate border and PNS neurons derived from this domain may not be innovations of vertebrates, but are rather ancient programmes for neuroectodermal patterning [122]. Under this model, the evolution of a neural crest GRN would have involved the integration of a downstream neural crest specification and migration module (e.g. SoxE, FoxD3, Tfap2a, Id, Snail).

Similar to the example of lateral neural borders, there is little evidence that the mechanisms of long-distance cell migration are unique to neural crest cells. Metazoans as diverse as sponges, diploblasts, annelids, molluscs, arthropods and deuterostomes all produce cells that undergo EMT and migrate. In fact, a potential synapomorphy of metazoans is the presence of mesenchyme and the ability of some cells to undergo EMTs and migrate during development. The widespread use of EMTs and cell migration seems to be underpinned by common molecular and cellular mechanisms as well. Cell migration in most metazoan

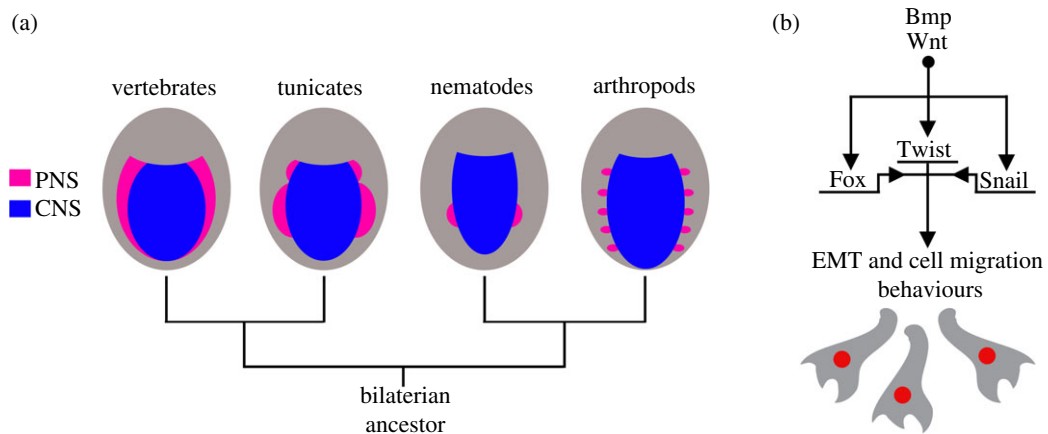

**Figure 3.** Some molecular, cellular and developmental hallmarks of neural crest cells long pre-date vertebrate and chordate origins. (*a*) Views of bilaterian gastrulae showing similar development of the neural plate border domain in vertebrates that generate most PNS sensory neurons and lateral neural borders in invertebrates that do the same. Both neural plate border and lateral neural border regions also express homologues of Msx, Pax, Zic and Nkx transcription factors. (*b*) Most metazoans use a common set of signalling molecules (Bmp and Wnt homologues) and transcription factors (Forkhead/Fox, Twist, Snail homologues) to initiate EMT and cell migration behaviours. (*a*) Modelled after [120].

embryos, including that of neural crest cells, involves evolutionarily conserved signalling inputs (Bmps, Wnts) that activate expression of pro-EMT transcription factors such as Twist, Fox and Snail which in turn modulate batteries of genes involved in intercellular adhesion and reconfiguration of the cytoskeleton (figure 3*b*). Thus, similar to the establishment of the neural plate border, neural crest cells share with many other cell types the genetic machinery for migration.

These observations together suggest that patterning of the lateral neuroectoderm into a PNS and production of cells that can migrate throughout the embryo long pre-date the advent of chordates and vertebrates and are therefore not exclusive to neural crest cells. What this points to is a scenario in which early chordates probably inherited these features from deep within the bilaterian tree. It is these features (lateral neural border with migratory cells) that may have served as a developmental blueprint for the evolution of neural crest-like cells that would appear in the last common ancestor of tunicates and vertebrates, after their split from the cephalochordate lineage around 600 Ma (figure 4).

## 3.2. Neural crest-like cells in invertebrate chordates: evolution of cells with limited migration and differentiation potential

Based on multiple lines of evidence, early chordates probably did not have neural crest or neural crest-like cells. Rather, the first chordates probably inherited a neural plate border (or lateral neural border) involved in medial–lateral patterning of the embryonic neuroectoderm and production of PNS neurons (figure 4), as well as the potential to generate sensory neurons from the ventral epidermis (a feature probably lost in vertebrates [120]). With the evolution of the lineage leading to tunicates and vertebrates (olfactores), we see for the first time cells that have a characteristic neural crest 'signature'. As described in §2.1, several hypotheses have been proposed to explain how changes in gene number and gene regulation might account for the origin of neural crest and neural crest-like cells. Is there any evidence to support these hypotheses?

It has been suggested that gene duplications were a driving force in neural crest evolution (see §2.1). However, the two rounds of genome duplication thought to have enabled sophistication of GRNs in vertebrate genomes cannot account for the appearance of neural crest-like cells in tunicates [95,98,123–126]. These animals show no evidence of having undergone genome duplications and in fact have probably experienced genome loss and contraction [127–129]. Rather, a major feature that distinguishes tunicates from cephalochordates with respect to neural crest-like cells is the expression of regulatory genes in the tunicate neural plate border such as Snail, Pax3/7, Zic, Msx and FoxD (figure 4). This provides evidence that in the lineage leading to tunicates and vertebrates, changes in *cis*-regulatory sequences were sufficient to integrate these genes within the neural plate border without the need for gene/genome duplications. Similarly, neural crest-like cells can modulate intercellular adhesion proteins requisite for delamination, EMT and migration [119], a result which suggests that the novel *cis*-regulation of epithelial versus mesenchymal gene batteries within the neural plate border occurred in the common ancestor of tunicates and vertebrates, independent of gene duplication events.

Although neural crest-like cells in tunicates are similar to neural crest cells, there are also notable differences. Importantly, neural crest-like cells can only generate single-cell types (pigment, otolith, ocellus, neurons). Additionally, none of these cells seem to migrate far from their site of origin, unlike the case in vertebrates in which neural crest cells engage in long-range, directed migration throughout the head and trunk. These features suggest an origin of a neural crest-like population in the last common ancestor of tunicates and vertebrates derived from the neural plate border, but one capable of giving rise to cells of limited potency and migratory capacity (figure 4).

## 3.3. Neural crest in early vertebrates: integration of multipotency with long-range and directed cell migration

In the previous sections, we have described an inventory of molecular, cellular and genetic features of the neural crest

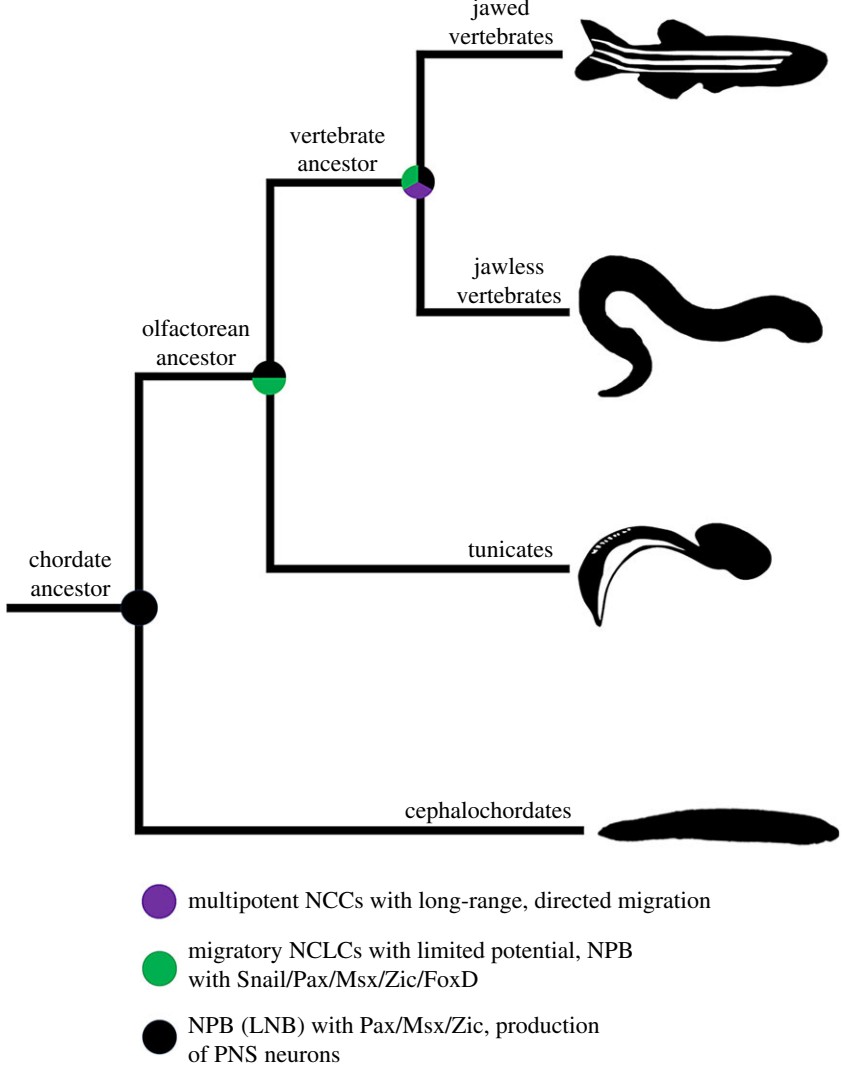

**Figure 4.** Model for the stepwise evolution of neural crest cells from ancestral precursors mapped onto a chordate phylogeny.

that long pre-date the evolutionary origins of this cell type. Under our scenario, it seems likely that early chordates had a neural plate border domain that produced PNS sensory neurons, and in the chordate ancestors of tunicates and vertebrates, a population of neural crest-like precursor cells evolved. These neural crest-like cells probably originated within a neural plate border domain that expressed Snail, Pax3/7, Zic and FoxD, and were capable of migrating and generating neural crest derivatives such as sensory neurons and pigment cells. Much of what minimally defines the neural crest, then, can already be identified in invertebrate chordates. And yet, invertebrate neural crest-like cells are clearly not the same as vertebrate neural crest. There are still key differences that span this evolutionary gap. Perhaps the most important differences between these two cell types are: (i) potency and 'stem-ness' that enable the production of both ectomesenchyme (e.g. cartilage and bone) and non-ectomesenchyme (e.g. neurons, glia, pigment) and (ii) the capacity for long-distance and directed migration (figure 4).

Neural crest cells are multipotent stem cells. They can produce more of their own cell type, which are in turn capable of generating a wide range of differentiated cells such as cartilage, bone, tendon and connective tissue, neurons, glia, pigment, tooth primordia, parts of the heart and endocrine system, and more [8–11]. This is not the case for neural crest-like cells. Indeed, all of the neural crest-like

cells in tunicates seem to be unipotent, generating a small number of non-ectomesenchymal derivatives such as neurons (BTNs) and pigment cells (otolith, ocellus) [118,119]. This means that the first neural crest-like cells probably had limited potency and that one of the key evolutionary innovations in early vertebrates would have been the evolution of a GRN that endowed cells from the neural plate border with the ability to produce multiple cell types of both non-ectomesenchymal and ectomesenchymal origin. How this GRN was assembled, however, has remained elusive. Work in frog embryos has begun to shed light on the matter. These studies have revealed that many neural plate border and neural crest specifier genes are actually expressed much earlier in development in the pluripotent animal pole cells of the frog blastula, alongside the core Sox–Oct–Myc–Vent pluripotency axis [130]. Moreover, the neural crest factors Snail1 and Sox5 regulate the blastula stem cell programme, suggesting that neural crest and pluripotent blastula cells share a common regulatory programme [130]. Finally, a SoxB1-to-SoxE switch during the transition from pluripotent blastula cells to neural crest stem cells might explain mechanistically how neural crest cells gradually acquire their stem cell state from an earlier pluripotent cell population [131]. If this model is correct, then it would provide evidence that a heterochronic shift, involving retention of a partial pluripotent state from blastula cells into neural

royalsocietypublishing.org/journal/rsob Open Biol. 10: 190285

crest, would have led to the evolution of the neural crest stem cell programme in early vertebrates [130,131]. The evolution of this stem cell regulatory state was particularly important for the evolution of ectomesenchymal cell types, such as cartilage and bone that comprise the vertebrate 'new head', a feature shared among both fossil forms and extant species [13,113,132–137]. These new ectomesenchyme-derived features enabled the evolution of a robust craniofacial skeleton to support the brain and sensory structures in early vertebrates, and eventually, articulated jaws with teeth for active predation in stem- and crown-group jawed vertebrates [13,132,138].

A second key feature of migratory crest cells in vertebrates that is apparently lacking in neural crest-like cells is their ability to embark on long-range migration throughout the embryo in a directed fashion. Compared with neural crest cells, both a9.49-derived cells and BTN precursors in tunicates migrate only a short distance from their site of origin before undergoing differentiation [118,119]. However, forced expression of Twist, a known regulator of EMT and cell migration in neural crest cells, was found to induce long-range, neural crest-like migration of multiple a9.49-derived cells into the larval tunic. This result suggests that the co-option of possibly a single transcription factor capable of regulating an EMT-type process would have been sufficient to produce cell migration behaviours reminiscent of neural crest cells [118].

Related to the long-range migration of neural crest cells in vertebrate embryos is the deployment of cellular communication systems (e.g. Sema/Nrp, Robo/Slit) that guide migratory crest and instruct their formation into specific morphological structures such as the head skeleton and PNS [139–141]. Although orthologues of some of these pathways have been identified in invertebrate chordates [142], their co-option by neural crest cells was instrumental in shaping vertebrate novelties [143]. This indicates that together with long-distance migration and multipotency, a key evolutionary step in the origin of neural crest cells was the ability of this population to migrate along defined routes throughout the embryo and to be shaped by cellular communication systems into three-dimensional structures.

In summary, our comparative analysis suggests that neural crest cells in the first vertebrates evolved at least three specific features—multipotency, long-range migration and cellular communication systems for guidance—that distinguished this cell population from neural crest-like cells in invertebrate chordates (figure 4). Under this model, retention of a pluripotency-like programme coupled with co-option of one or a few EMT regulators (e.g. Twist) may have been sufficient to produce a migratory and multipotent cell population. However, the co-option of intercellular communications systems would have been another importance advance in shaping these migratory stem cells into many of the morphological novelties that define the vertebrate clade.

Data accessibility. This article has no additional data.
Authors' contributions. J.R.Y. and D.W.M. wrote the article.
Competing interests. We declare we have no competing interests.
Funding. J.R.Y and D.W.M received internal funding support from the University of Oklahoma.

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
