## [Reviewer comments · Open Biology]

Review History

RSOB-19-0285.R0 (Original submission)

Review form: Reviewer 1

Recommendation

Accept with minor revision (please list in comments)

Do you have any ethical concerns with this paper?

No

Comments to the Author

This is a nice review covering an important topic. It is well written and clear. The figures are simple but effective. There is a nice summary of the current state of knowledge regarding the molecular and genetic understanding of neural crest development. There is a good clear discussion of the current state of the search for neural crest precursors in non-vertebrate chordates. I have only some minor points for consideration..

The most important one is that there is little discussion of the ectomesenchymal capacities of the neural crest - their ability to generate dento-skeletal tissue. This is important as it is this that is the defining feature of vertebrae neural crest cells. A link here to the fossil data would be useful.

Other minor points -

The last sentence of the abstract - "In stem lineages, a heterochronic shift of pluripotency coupled with long-range and directed migration led to the transition from neural crest-like cells in chordates to multipotent migratory neural crest in the first vertebrates."

This needs clarity. I can guess what the authors are driving at but it far from clear.

Part 2.2 Tunicates, it is stated that "These "neural crest-like cells" also expressed neural crest markers such as Zic and HNK1 [112-114]. Although subsequent studies revealed the expression of additional neural crest regulatory genes in these cells, the lineage (a7.6) they were derived from turned out to be mesoderm, rather than neural plate border, raising doubts about their homology with neural crest [112-114]."

This is only true of the studies using *Ciona* this statement does not apply to those involving *Ecteinascidia* ref 114 and thus misrepresents this paper.

Review form: Reviewer 2

Recommendation

Accept with minor revision (please list in comments)

Do you have any ethical concerns with this paper?

No

Comments to the Author

In the present manuscript York, J. and McCauley D. review present knowledge about the origin and evolution of a very important vertebrate cell-type, the neural crest cells. The manuscript is correctly written and collects up-to-date information, giving a general view about stepwise evolution of these migratory cell population. For these reasons I accept this manuscript for publication in Open Biology after correction of some minor comments that I describe below in order of appearance in the text.

Abstract: "...led to the transition from neural crest-like cells in chordates to multipotent migratory neural crest in the first vertebrates." Since vertebrates are also chordates, I would change this phrase to: "...led to the transition from neural crest-like cells in invertebrate-chordates to multipotent migratory neural crest in the first vertebrates"

Part 2. 1: In the two first lines, in order to be coherent change "...include the cephalochordates, represented by amphioxus, and the tunicates (also known as urochordates)", by "...include the cephalochordates, represented by amphioxus, and the urochordates (also known as tunicates)"

Part 2.1 page 5: Cephalochordates are not "represented" by amphioxus. Amphioxus is the popular name for this invertebrate-chordate subphylum, so cephalochordates ARE amphioxus. I suggest to change to "cephalochordates (i.e. amphioxus). Moreover, the present phylogenetic distribution within chordates, placing cephalochordates as the most basally-divergent lineage, was not based in whole genome molecular studies, but on concatenations of a high number of gene sequences. The reference given here for this new phylogenetic distribution (Delsuc et al 2006) is correct but this paper placed cephalochordates as sister group of echinoderms, and the first paper where tunicates appear as sister group of vertebrates keeping the monophyly of chordates should also be cited here (the paper is Bourlat, S. J et al (2006). Nature 444(7115): 85-88.)

Part 2.1, page 6: When describing the fate of duplicated genes, please cite the original paper where the three possible fates following a gene duplication event were first described. This reference is

Force, A., et al (1999). *Genetics* 151(4): 1531-1545. Moreover, a recent publication adds a new explanation for retention of duplicated genes which is “cooperation”, and it would be interesting to cite this paper here also with a few words explaining this fourth possibility (Chapal, M., et al (2019). *PLOS Biology* 17(11): e3000289.)

Part 2.1, page 7: The authors explain here that previous findings suggest that “...the amphioxus regulatory elements lack the regulatory sites to mediate expression in the neural crest, which likely evolved in early vertebrates.” Moreover, the authors use a recent publication (Marletaz et al 2019, *Nature*; attention concerning this reference it misses most of the authors in the reference list) to support this assumption by saying “Recent comparisons of whole-genome regulatory landscapes between vertebrates and amphioxus have arrived at similar conclusions”. However, what this paper shows is, that there is a regulatory complexification in vertebrates compared with amphioxus in terms of number of enhancers per gene, not in terms of binding sites in these enhancers. In addition, the authors add “Compared to amphioxus, the cis-regulatory elements in vertebrate neural crest paralogs are richer and more complex with respect to transcription factor binding sites and have achieved much greater specialization and precision in spatial-temporal expression compared to the ancestral chordate condition” and to my knowledge there is no specific view on NC gene paralogs in Marletaz et al. Finally, the authors state in the last phrase of this part that “much of the complexity of cis-regulatory control in vertebrate genomes may be attributable largely to gene and/or genome duplication” but the authors do not explain why. If the authors want to attribute the complexification of cis-regulation to the 2R, they should explain why following the conclusions shown in Marletaz et al. The regulatory complexity of vertebrates following Marletaz et al results show that gene families where more paralogs have been retained following the 2R, show a higher number of enhancers, so gene families which kept a single copy gene have a similar number of enhancers per gene as amphioxus but gene families with four paralogs have a higher number of enhancers per gene compared with their orthologue in amphioxus. So, if the authors want to a

Part 3.1, page 9: please change “An adequate answer to this question has become elusive, given the discovery of neural crest-like cells in invertebrates.” to “An adequate answer to this question has become elusive, given the discovery of neural crest-like cells in invertebrate-chordates” (or ...in tunicates).

Part 3.2: The authors explain that the first chordates inherited a neural plate border which produces PNS neurons. I would also add somewhere, the fact that invertebrate chordates also produce PNS neurons from the ventral epidermis (something that vertebrates lost).

Part 3.2: The authors say that several hypotheses have been proposed to explain how changes in gene number and gene regulation might account for the origin of NC. While Marletaz et al show a complexification of gene regulation following the 2R, nothing accounts for the possibility of a higher number of genes played a role in the appearance of any given tissue (including NC). If the authors have any other study showing the new genes coming from duplication events played a role in the evolution of NC, please cite here.

Part 3.3: The authors state that *Sema3F/Nrp* signalling pathway is unique to vertebrates and give an auto-reference of 2018 in *Development* which in turn cites a 2006 work which studied the phylogeny of this family using only sequences from vertebrates, *Drosophila*, *C. elegans* and virus (Yazdani and Terman, 2006). Logically, in this study the *Sema3* subfamily is vertebrate-specific since it does not exist in *Drosophila* nor *C. elegans*. However, in more recent studies (see for example Junqueira Alves, et al (2019). *Sci Rep* 9(1): 1970), it is clearly observed that what is specific to vertebrates is the duplication due to the 2R that gives rise to *Sema3* and *Sema4* paralogues, while amphioxus has a unique "preduplicated" gene that has been called *Sema6*, although it should be called *Sema3/4*. All this to say that the statement that *Sema3* is specific to vertebrates is not true since in amphioxus there is a gene (*Sema3/4*) that is coorthologue of these two vertebrate genes.

Decision letter (RSOB-19-0285.R0)

20-Dec-2019

Dear Dr McCauley,

We are pleased to inform you that your manuscript RSOB-19-0285 entitled "The origin and evolution of vertebrate neural crest cells" has been accepted by the Editor for publication in Open Biology. The reviewer(s) have recommended publication, but also suggest some minor revisions to your manuscript. Therefore, we invite you to respond to the reviewer(s)' comments and revise your manuscript.

Please submit the revised version of your manuscript within 21 days. If you do not think you will be able to meet this date please let us know immediately and we can extend this deadline for you.

- 1) A text file of the manuscript (doc, txt, rtf or tex), including the references, tables (including captions) and figure captions. Please remove any tracked changes from the text before submission. PDF files are not an accepted format for the "Main Document".
- 2) A separate electronic file of each figure (tiff, EPS or print-quality PDF preferred). The format should be produced directly from original creation package, or original software format. Please note that PowerPoint files are not accepted.
- 3) Electronic supplementary material: this should be contained in a separate file from the main text and meet our ESM criteria (see <http://royalsocietypublishing.org/instructions-authors#question5>). All supplementary materials accompanying an accepted article will be treated as in their final form. They will be published alongside the paper on the journal website and posted on the online figshare repository. Files on figshare will be made available approximately one week before the accompanying article so that the supplementary material can be attributed a unique DOI.

Online supplementary material will also carry the title and description provided during submission, so please ensure these are accurate and informative. Note that the Royal Society will not edit or typeset supplementary material and it will be hosted as provided. Please ensure that the supplementary material includes the paper details (authors, title, journal name, article DOI). Your article DOI will be 10.1098/rsob.2016[last 4 digits of e.g. 10.1098/rsob.20160049].

4) A media summary: a short non-technical summary (up to 100 words) of the key findings/importance of your manuscript. Please try to write in simple English, avoid jargon, explain the importance of the topic, outline the main implications and describe why this topic is newsworthy.

Images

Data-Sharing

It is a condition of publication that data supporting your paper are made available. Data should be made available either in the electronic supplementary material or through an appropriate repository. Details of how to access data should be included in your paper. Please see <http://royalsocietypublishing.org/site/authors/policy.xhtml#question6> for more details.

Data accessibility section

Sincerely,

The Open Biology Team

<mailto:openbiology@royalsociety.org>

Reviewer(s)' Comments to Author:

Referee: 1

Comments to the Author(s)

This is a nice review covering an important topic. It is well written and clear. The figures are simple but effective. There is a nice summary of the current state of knowledge regarding the molecular and genetic understanding of neural crest development. There is a good clear discussion of the current state of the search for neural crest precursors in non-vertebrate chordates. I have only some minor points for consideration..

The most important one is that there is little discussion of the ectomesenchymal capacities of the neural crest - their ability to generate dento-skeletal tissue. This is important as it is this that is the defining feature of vertebrae neural crest cells. A link here to the fossil data would be useful.

Other minor points -

The last sentence of the abstract - "In stem lineages, a heterochronic shift of pluripotency coupled with long-range and directed migration led to the transition from neural crest-like cells in chordates to multipotent migratory neural crest in the first vertebrates."

This needs clarity. I can guess what the authors are driving at but it far from clear.

Part 2.2 Tunicates, it is stated that "These "neural crest-like cells" also expressed neural crest markers such as Zic and HNK1 [112-114]. Although subsequent studies revealed the expression of additional neural crest regulatory genes in these cells, the lineage (a7.6) they were derived from turned out to be mesoderm, rather than neural plate border, raising doubts about their homology with neural crest [112-114]."

This is only true of the studies using *Ciona* this statement does not apply to those involving *Ecteinascidia* ref 114 and thus misrepresents this paper.

Referee: 2

Comments to the Author(s)

In the present manuscript York, J. and McCauley D. review present knowledge about the origin and evolution of a very important vertebrate cell-type, the neural crest cells. The manuscript is correctly written and collects up-to-date information, giving a general view about stepwise evolution of these migratory cell population. For these reasons I accept this manuscript for publication in *Open Biology* after correction of some minor comments that I describe below in order of appearance in the text.

Abstract: "...led to the transition from neural crest-like cells in chordates to multipotent migratory neural crest in the first vertebrates." Since vertebrates are also chordates, I would change this phrase to: "...led to the transition from neural crest-like cells in invertebrate-chordates to multipotent migratory neural crest in the first vertebrates"

Part 2. 1: In the two first lines, in order to be coherent change "...include the cephalochordates, represented by amphioxus, and the tunicates (also known as urochordates)", by "...include the cephalochordates, represented by amphioxus, and the urochordates (also known as tunicates)"

Part 2.1 page 5: Cephalochordates are not "represented" by amphioxus. Amphioxus is the popular name for this invertebrate-chordate subphylum, so cephalochordates ARE amphioxus. I suggest to change to "cephalochordates (i.e. amphioxus). Moreover, the present phylogenetic distribution within chordates, placing cephalochordates as the most basally-divergent lineage, was not based in whole genome molecular studies, but on concatenations of a high number of gene sequences. The reference given here for this new phylogenetic distribution (Delsuc et al 2006) is correct but this paper placed cephalochordates as sister group of echinoderms, and the first paper where tunicates appear as sister group of vertebrates keeping the monophyly of chordates should also be cited here (the paper is Bourlat, S. J et al (2006). *Nature* 444(7115): 85-88.)

Part 2.1, page 6: When describing the fate of duplicated genes, please cite the original paper where the three possible fates following a gene duplication event were first described. This reference is Force, A., et al (1999). *Genetics* 151(4): 1531-1545. Moreover, a recent publication adds a new explanation for retention of duplicated genes which is "cooperation", and it would be interesting to cite this paper here also with a few words explaining this fourth possibility (Chapal, M., et al (2019). *PLOS Biology* 17(11): e3000289.)

Part 2.1, page 7: The authors explain here that previous findings suggest that "...the amphioxus regulatory elements lack the regulatory sites to mediate expression in the neural crest, which likely evolved in early vertebrates." Moreover, the authors use a recent publication (Marletaz et al 2019, *Nature*; attention concerning this reference it misses most of the authors in the reference list) to support this assumption by saying "Recent comparisons of whole-genome regulatory landscapes between vertebrates and amphioxus have arrived at similar conclusions". However, what this paper shows is, that there is a regulatory complexification in vertebrates compared with amphioxus in terms of number of enhancers per gene, not in terms of binding sites in these enhancers. In addition, the authors add "Compared to amphioxus, the cis-

regulatory elements in vertebrate neural crest paralogs are richer and more complex with respect to transcription factor binding sites and have achieved much greater specialization and precision in spatial-temporal expression compared to the ancestral chordate condition" and to my knowledge there is no specific view on NC gene paralogs in Marletaz et al. Finally, the authors state in the last phrase of this part that "much of the complexity of cis-regulatory control in vertebrate genomes may be attributable largely to gene and/or genome duplication" but the authors do not explain why. If the authors want to attribute the complexification of cis-regulation to the 2R, they should explain why following the conclusions shown in Marletaz et al. The regulatory complexity of vertebrates following Marletaz et al results show that gene families where more paralogs have been retained following the 2R, show a higher number of enhancers, so gene families which kept a single copy gene have a similar number of enhancers per gene as amphioxus but gene families with four paralogs have a higher number of enhancers per gene compared with their orthologue in amphioxus. So, if the authors want to a

Part 3.1, page 9: please change "An adequate answer to this question has become elusive, given the discovery of neural crest-like cells in invertebrates." to "An adequate answer to this question has become elusive, given the discovery of neural crest-like cells in invertebrate-chordates" (or ...in tunicates).

Part 3.2: The authors explain that the first chordates inherited a neural plate border which produces PNS neurons. I would also add somewhere, the fact that invertebrate chordates also produce PNS neurons from the ventral epidermis (something that vertebrates lost).

Part 3.2: The authors say that several hypotheses have been proposed to explain how changes in gene number and gene regulation might account for the origin of NC. While Marletaz et al show a complexification of gene regulation following the 2R, nothing accounts for the possibility of a higher number of genes played a role in the appearance of any given tissue (including NC). If the authors have any other study showing the new genes coming from duplication events played a role in the evolution of NC, please cite here.

Part 3.3: The authors state that Sema3F/Nrp signalling pathway is unique to vertebrates and give an auto-reference of 2018 in Development which in turn cites a 2006 work which studied the phylogeny of this family using only sequences from vertebrates, drosophila, C. elegans and virus (Yazdani and Terman, 2006). Logically, in this study the Sema3 subfamily is vertebrae-specific since it does not exist in Drosophila nor C. elegans. However, in more recent studies (see for example Junqueira Alves, et al (2019). Sci Rep 9(1): 1970), it is clearly observed that what is specific to vertebrates is the duplication due to the 2R that gives rise to Sema3 and Sema4 paralogues, while amphioxus has a unique "preduplicated" gene that has been called Sema6, although it should be called Sema3/4. All this to say that the statement that Sema3 is specific to vertebrates is not true since in amphioxus there is a gene (Sema3/4) that is coorthologue of these two vertebrate genes.

Author's Response to Decision Letter for (RSOB-19-0285.R0)

See Appendix A.

Decision letter (RSOB-19-0285.R1)

06-Jan-2020

Dear Dr McCauley

We are pleased to inform you that your manuscript entitled "The origin and evolution of vertebrate neural crest cells" has been accepted by the Editor for publication in Open Biology.

Sincerely,

The Open Biology Team
mailto: openbiology@royalsociety.org

Appendix A

From: Open Biology <onbehalf@manuscriptcentral.com>
Sent: Friday, December 20, 2019 10:27 AM
To: McCauley, David W.
Subject: Open Biology - Decision on Manuscript RSOB-19-0285

20-Dec-2019

Dear Dr McCauley,

We are pleased to inform you that your manuscript RSOB-19-0285 entitled "The origin and evolution of vertebrate neural crest cells" has been accepted by the Editor for publication in Open Biology. The reviewer(s) have recommended publication, but also suggest some minor revisions to your manuscript. Therefore, we invite you to respond to the reviewer(s)' comments and revise your manuscript.

Please submit the revised version of your manuscript within 21 days. If you do not think you will be able to meet this date please let us know immediately and we can extend this deadline for you.

- 1) A text file of the manuscript (doc, txt, rtf or tex), including the references, tables (including captions) and figure captions. Please remove any tracked changes from the text before submission. PDF files are not an accepted format for the "Main Document".
- 2) A separate electronic file of each figure (tiff, EPS or print-quality PDF preferred). The format should be produced directly from original creation package, or original software format. Please note that PowerPoint files are not accepted.
- 3) Electronic supplementary material: this should be contained in a separate file from the main text and meet our ESM criteria (see <http://royalsocietypublishing.org/instructions-authors#question5>). All supplementary materials accompanying an accepted article will be treated as in their final form. They will be published alongside the paper on the journal website and posted on the online figshare repository. Files on figshare will be made available

approximately one week before the accompanying article so that the supplementary material can be attributed a unique DOI.

Online supplementary material will also carry the title and description provided during submission, so please ensure these are accurate and informative. Note that the Royal Society will not edit or typeset supplementary material and it will be hosted as provided. Please ensure that the supplementary material includes the paper details (authors, title, journal name, article DOI). Your article DOI will be 10.1098/rsob.2016[last 4 digits of e.g. 10.1098/rsob.20160049].

4) A media summary: a short non-technical summary (up to 100 words) of the key findings/importance of your manuscript. Please try to write in simple English, avoid jargon, explain the importance of the topic, outline the main implications and describe why this topic is newsworthy.

Images

Data-Sharing

It is a condition of publication that data supporting your paper are made available. Data should be made available either in the electronic supplementary material or through an appropriate repository. Details of how to access data should be included in your paper. Please see <http://royalsocietypublishing.org/site/authors/policy.xhtml#question6> for more details.

Data accessibility section

Sincerely,

The Open Biology Team
<mailto:openbiology@royalsociety.org>

Dear Editors and Referees,

Thank you for the constructive reviews of our manuscript entitled "*The origin and evolution of vertebrate neural crest cells*" (RSOB-19-0285) submitted for publication consideration in *Open Biology*. We have submitted a revised version of the manuscript with the same title that incorporates the helpful and important suggestions from all referees. We have also prepared a detailed, itemized description of each of the changes made to the revised manuscript where we address referee comments. We include referee comments in the response letter in non-bold italics, and our responses and descriptions of changes to the manuscript in bold. In their original comments, the referees referred to sections and/or paragraphs rather than line numbers. In our responses, we keep to this original format and refer to these same sections/paragraphs/page numbers for our revised manuscript.

Reviewer(s)' Comments to Author:

Referee: 1

Comments to the Author(s)

This is a nice review covering an important topic. It is well written and clear. The figures are simple but effective. There is a nice summary of the current state of knowledge regarding the molecular and genetic understanding of neural crest development. There is a good clear discussion of the current state of the search for neural crest precursors in non-vertebrate chordates. I have only some minor points for consideration..

The most important one is that there is little discussion of the ectomesenchymal capacities of the neural crest - their ability to generate dento-skeletal tissue. This is important as it is this that is the defining feature of vertebrae neural crest cells. A link here to the fossil data would be useful.

Response: This is a great suggestion. We now discuss the significance of vertebrate neural crest giving rise to ectomesenchyme, and in particular, the evolution of the craniofacial skeleton (first two paragraphs of section 3.3), where we describe the evolution of neural crest multipotency. We think it best to include this new text here because the ability of neural crest to generate ectomesenchyme is linked directly to the evolution of multipotency and the ability to produce new cell types. We also briefly discuss how this is linked to the "new head" hypothesis and include relevant literature from the fossil record.

Other minor points -

The last sentence of the abstract - "In stem lineages, a heterochronic shift of pluripotency coupled with long-range and directed migration led to the transition from neural crest-like cells in chordates to multipotent migratory neural crest in the first vertebrates."

This needs clarity. I can guess what the authors are driving at but it far from clear.

Response: We have modified this sentence for clarity in the last sentence of the abstract.

Part 2.2 Tunicates, it is stated that "These "neural crest-like cells" also expressed neural crest

markers such as Zic and HNK1 [112-114]. Although subsequent studies revealed the expression of additional neural crest regulatory genes in these cells, the lineage (a7.6) they were derived from turned out to be mesoderm, rather than neural plate border, raising doubts about their homology with neural crest [112-114]."

This is only true of the studies using *Ciona* this statement does not apply to those involving *Ecteinascidia* ref 114 and thus misrepresents this paper.

Response: We have modified this text in the first paragraph of section 2.2 to indicate that dissimilarities in these trunk lateral cell/NCLCs and vertebrate neural crest is observed primarily in *Ciona* species.

Referee: 2

Comments to the Author(s)

In the present manuscript York, J. and McCauley D. review present knowledge about the origin and evolution of a very important vertebrate cell-type, the neural crest cells. The manuscript is correctly written and collects up-to-date information, giving a general view about stepwise evolution of these migratory cell population. For these reasons I accept this manuscript for publication in Open Biology after correction of some minor comments that I describe below in order of appearance in the text.

Abstract: "...led to the transition from neural crest-like cells in chordates to multipotent migratory neural crest in the first vertebrates." Since vertebrates are also chordates, I would change this phrase to: "...led to the transition from neural crest-like cells in invertebrate-chordates to multipotent migratory neural crest in the first vertebrates"

Response: Done. This sentence of the abstract as also been revised slightly for clarity in response to Referee 1.

Part 2. 1: In the two first lines, in order to be coherent change "...include the cephalochordates, represented by amphioxus, and the tunicates (also known as urochordates)", by "...include the cephalochordates, represented by amphioxus, and the urochordates (also known as tunicates)"

Response: This first sentence of section 2.1 has been revised accordingly, but please recognize that this same sentence has been modified per this Referee's next comment.

Part 2.1 page 5: Cephalochordates are not "represented" by amphioxus. Amphioxus is the popular name for this invertebrate-chordate subphylum, so cephalochordates ARE amphioxus. I suggest to change to "cephalochordates (i.e. amphioxus). Moreover, the present phylogenetic distribution within chordates, placing cephalochordates as the most basally-divergent lineage, was not based in whole genome molecular studies, but on concatemers of a high number of gene sequences. The reference given here for this new phylogenetic distribution (Delsuc et al 2006) is correct but this paper placed cephalochordates as sister group of echinoderms, and the first paper were tunicates appear as sister group of vertebrates keeping the monophyly of chordates should also be cited here (the paper is Bourlat, S. J et al (2006). Nature 444(7115): 85-88.)

Response: We have made changes to the second half of the first paragraph in section 2.1 to indicate that amphioxus are cephalochordates and that the first studies demonstrating

a vertebrate+tunicate sister group were molecular phylogenetic studies, not whole-genome analyses. We have included the Bourlat et al. (2006) Nature reference.

Part 2.1, page 6: When describing the fate of duplicated genes, please cite the original paper where the three possible fates following a gene duplication event were first described. This reference is Force, A., et al (1999). Genetics 151(4): 1531-1545. Moreover, a recent publication adds a new explanation for retention of duplicated genes which is “cooperation”, and it would be interesting to cite this paper here also with a few words explaining this fourth possibility (Chapal, M., et al (2019). PLOS Biology 17(11): e3000289.)

Response: We have included the original Force et al. 1999 (Genetics) reference to the second paragraph of section 2.1 (reference number 97). We also include a brief description of “cooperation” as a fourth possible result of gene duplication (end of second paragraph in section 2.1)

Part 2.1, page 7: The authors explain here that previous findings suggest that “...the amphioxus regulatory elements lack the regulatory sites to mediate expression in the neural crest, which likely evolved in early vertebrates.” Moreover, the authors use a recent publication (Marletaz et al 2019, Nature; attention concerning this reference it misses most of the authors in the reference list) to support this assumption by saying “Recent comparisons of whole-genome regulatory landscapes between vertebrates and amphioxus have arrived at similar conclusions”. However, what this paper shows is, that there is a regulatory complexification in vertebrates compared with amphioxus in terms of number of enhancers per gene, not in terms of binding sites in these enhancers. In addition, the authors add “Compared to amphioxus, the cis-regulatory elements in vertebrate neural crest paralogs are richer and more complex with respect to transcription factor binding sites and have achieved much greater specialization and precision in spatial-temporal expression compared to the ancestral chordate condition” and to my knowledge there is no specific view on NC gene paralogs in Marletaz et al. Finally, the authors state in the last phrase of this part that “much of the complexity of cis-regulatory control in vertebrate genomes may be attributable largely to gene and/or genome duplication” but the authors do not explain why. If the authors want to attribute the complexification of cis-regulation to the 2R, they should explain why following the conclusions shown in Marletaz et al. The regulatory complexity of vertebrates following Marletaz et al results show that gene families where more paralogs have been retained following the 2R, show a higher number of enhancers, so gene families which kept a single copy gene have a similar number of enhancers per gene as amphioxus but gene families with four paralogs have a higher number of enhancers per gene compared with their orthologue in amphioxus. So, if the authors want to a

Response: We have modified this text to indicate that the Marletaz paper describes increased regulatory complexity in terms of the number of enhancers and removed any specific reference of this paper to neural crest regulatory complexity (last paragraph of section 2.1). We have also added text indicating that post-duplication regulatory complexity gained in vertebrate genomes may be the result of one or more paralogs of these regulatory regions having acquiring novel enhancers because other paralogs would have still been able to perform the ancestral regulatory function(s) (last paragraph of section 2.1).

Part 3.1, page 9: please change “An adequate answer to this question has become elusive, given the discovery of neural crest-like cells in invertebrates.” to “An adequate answer to this question has become elusive, given the discovery of neural crest-like cells in invertebrate-chordates” (or ...in tunicates).

Response: Done (first sentence of first paragraph in section 3.1).

Part 3.2: The authors explain that the first chordates inherited a neural plate border which produces PNS neurons. I would also add somewhere, the fact that invertebrate chordates also produce PNS neurons from the ventral epidermis (something that vertebrates lost).

Response: Done (first paragraph of section 3.2).

Part 3.2: The authors say that several hypotheses have been proposed to explain how changes in gene number and gene regulation might account for the origin of NC. While Marletaz et al show a complexification of gene regulation following the 2R, nothing accounts for the possibility of a higher number of genes played a role in the appearance of any given tissue (including NC). If the authors have any other study showing the new genes coming from duplication events played a role in the evolution of NC, please cite here.

Response: We agree with the referee's comment. As we describe in our text here, it seems unlikely that 2R gene duplication can account for the evolution of neural crest cells, given that tunicates, which have not experienced these duplication events, nonetheless have cells very similar to neural crest. As such, we are unaware of studies directly linking 2R, or any other widespread gene duplication events to the evolution of the neural crest regulatory network and have not made any changes to this text.

*Part 3.3: The authors state that *Sema3F/Nrp* signalling pathway is unique to vertebrates and give an auto-reference of 2018 in *Development* which in turn cites a 2006 work which studied the phylogeny of this family using only sequences from vertebrates, *Drosophila*, *C. elegans* and virus (Yazdani and Terman, 2006). Logically, in this study the *Sema3* subfamily is vertebrate-specific since it does not exist in *Drosophila* nor *C. elegans*. However, in more recent studies (see for example Junqueira Alves, et al (2019). *Sci Rep* 9(1): 1970), it is clearly observed that what is specific to vertebrates is the duplication due to the 2R that gives rise to *Sema3* and *Sema4* paralogues, while amphioxus has a unique "preduplicated" gene that has been called *Sema6*, although it should be called *Sema3/4*. All this to say that the statement that *Sema3* is specific to vertebrates is not true since in amphioxus there is a gene (*Sema3/4*) that is coorthologue of these two vertebrate genes.*

Response: We have modified this text to indicate that, although recent studies have indeed found homologs of receptor-ligand guidance systems, such as *Sema3/4* in amphioxus, vertebrate neural crest cells have uniquely co-opted these guidance systems to organize large groups of migratory, multipotent cells down migratory routes and into specific morphological features that are unique to vertebrates (e.g., craniofacial skeleton) (third paragraph of section 3.3).